# A Comparison of Different Reagents Applicable for Destroying Halogenated Anionic Textile Dye Mordant Blue 9 in Polluted Aqueous Streams

**Barbora Kamenická and Tomáš Weidlich \***

Chemical Technology Group, Institute of Environmental and Chemical Engineering, Faculty of Chemical Technology, University of Pardubice, Studentska 573, 532 10 Pardubice, Czech Republic
* Correspondence: tomas.weidlich@upce.cz; Tel.: +420-46-603-8049

**Abstract:** This article aimed to compare the degradation efficiencies of different reactants applicable for the oxidative or reductive degradation of a chlorinated anionic azo dye, Mordant Blue 9 (MB9). In this article, the broadly applied Fenton oxidation process was optimized for the oxidative treatment of MB9, and the obtained results were compared with other innovative chemical reduction methods. In the reductive degradation of MB9, we compared the efficiencies of different reductive agents such as $Fe^0$ (ZVI), $Al^0$, the Raney Al-Ni alloy, $NaBH_4$, $NaBH_4/Na_2S_2O_5$, and other combinations of these reductants. The reductive methods aimed to reduce the azo bond together with the bound chlorine in the structure of MB9. The dechlorination of MB9 produces non-chlorinated aminophenols, which are more easily biodegradable in wastewater treatment plants (WWTPs) compared to their corresponding chlorinated aromatic compounds. The efficiencies of both the oxidative and reductive degradation processes were monitored by visible spectroscopy and determined based on the chemical oxygen demand (COD). The hydrodechlorination of MB9 to non-chlorinated products was expressed using the measurement of adsorbable organically bound halogens (AOXs) and controlled by LC–MS analyses. Optimally, 28 mol of $H_2SO_4$, 120 mol of $H_2O_2$, and 4 mol of $FeSO_4$ should be applied per one mol of dissolved MB9 dye for a practically complete oxidative degradation after 20 h of action. On the other hand, the application of the Al-Ni alloy/NaOH (100 mol of Al in the Al-Ni alloy + 100 mol of NaOH per one mol of MB9) proceeded smoothly and seven-times faster than the Fenton reaction, consumed similar quantities of reagents, and produced dechlorinated aminophenols. The cost of the Al-Ni alloy/NaOH-based method could be decreased significantly by applying a pretreatment with $Al^0/NaOH$ and a subsequent hydrodechlorination using smaller Al-Ni alloy doses. The homogeneous reduction accompanied by HDC using in situ produced $Na_2S_2O_4$ (by the action of $NaBH_4/Na_2S_2O_5$) was an effective, rapid, and simple treatment method. This reductive system consumed quantities of reagents that are almost twice as low (66 mol of $NaBH_4$ + 66 mol of $Na_2S_2O_5$ + 18 mol of $H_2SO_4$ per one mol of MB9) in comparison with the other oxidative/reductive systems and allowed the effective and fast degradation of MB9 accompanied by the effective removal of AOX. A comparison of the oxidative and reductive methods for chlorinated acid azo dye MB9 degradation showed that an innovative combination of reduction methods offers a smooth, simple, and efficient degradation and hydrodehalogenation of chlorinated textile MB9 dye.

**Keywords:** acid azo dye; Mordant Blue 9; chlorophenol; Fenton oxidation; degradation; hydrodehalogenation; hydrodechlorination; $NaBH_4$; $Na_2S_2O_5$; Raney Al-Ni alloy

## 1. Introduction

Azo dyes are the most important synthetic colorants and have been used for a wide spectrum of textile dyeing [1,2]. Acid dyes soluble in aqueous solutions are usually applied for dyeing natural fibers [3,4]. Mordant dyes derived from acid dyes have been widely used for the coloration of textile fibers such as wool, silk, polyester, cotton, and some

modified cellulose fibers [5–8]. One example of the mentioned mordant azo dyes is the acid dye Mordant Blue 9 (MB9, Figure 1). However, the effective treatment of dye baths and wastewater from dye production and utilization before they are released to the environment has received significant attention because of their potential toxic effects based on the fact that Mordant dyes are difficult to degrade [9].

**Figure 1.** The chemical structure of anionic azo dye Mordant Blue 9.

On average, the annual production of dyes exceeds $7 \times 10^5$ tons. However, in the textile industry, typically about 10% of the used dyes are discharged, with the effluent creating both environmental and economic issues due to the limited biodegradability of azo dyes, especially halogenated ones [10]. The occurrence of especially halogenated non-biodegradable azo dyes in wastewater not only deteriorates water aesthetically due to its coloration but also increases its chemical oxygen demand (COD) and adsorbable organic halogen content (AOX) [11].

Various removal methods have been developed to remove the dyes produced during the dyeing of textile fabrics from wastewater. However, the effective destruction of non-biodegradable azo dyes in effluents does not only comprise decolorization [12–17]. The practical removal of spent dyes in wastewater streams should be accompanied by the formation of easily biodegradable products or by the complete mineralization of the original dye to $CO_2$, $H_2O$, nitrate or $N_2$, sulphate, etc. [16,17].

Fenton-based advanced oxidation processes (AOPs) are among the most effective processes because of their relative ease of operation, low-cost facilities, and high removal efficiency [18].

Fenton oxidation degrades organic pollutants by applying hydroxyl radicals (HO·), which are generated from a hydrogen peroxide ($H_2O_2$) + ferrous sulfate reaction in an acidic medium [19]. However, the common application of Fenton processes is limited by several limitations, such as acidic pH conditions, iron-based sludge generation, and the large consumption of hydrogen peroxide [20].

The large consumption of $H_2O_2$ in Fenton oxidation is caused by the high reactivity (low stability) of the produced OH· radicals together with the easier degradability of the organic oxidation products compared to the starting pollutant [18]. In the case of chemical oxidation, a complete mineralization is typically necessary for the removal of toxic and non-biodegradable halogenated organic by-products such as AOX [13].

Some improvements in the utilization of the oxidant used in chemical oxidation processes provides the substitution of hydrogen peroxide with persulfate ($S_2O_8^{2-}$) [18].

The formation of the sulfate radical $\cdot SO_4^-$ from alkali metal persulfate is commonly catalyzed by the action of transition metal ions such as $Fe^{2+}$, $Co^{2+}$, etc. The sulfate radical works as a strong oxidant possessing a higher redox potential and an even higher span of its half-life stability compared with hydroxyl radicals [21–27].

However, the higher price of persulfate (800–1160 USD/t of $Na_2S_2O_8$ [28]) compared to hydrogen peroxide (350–450 USD/t of $H_2O_2$ [29]) and the secondary contamination of treated water with inorganic sulphates are the limitations of persulfate-based AOPs.

Compared with chemical oxidation, the chemical reduction method enables more selective reductive cleavage of azo or other simply reducible bonds ($NO_2$, etc.) [30]. Applying appropriate reductants, it facilitates the effective hydrodehalogenation of carbon-halogen

bonds that occur in many synthetic dyes (the reductive removal of AOX-producing inorganic halides) [31].

Several papers [32–35] have described the reductive degradation of azo dyes using zero-valent iron ($Fe^0$ or ZVI) or nano ZVI. However, ZVI is not able to effectively reduce carbon–halogen bonds in halogenated aromatic compounds [33].

As we recently published, the Raney Al-Ni alloy is an excellent hydrodechlorination (HDC) agent [30,31]. Hydrodehalogenation using Al-based alloys is based on dissolving aluminium in the sodium hydroxide solution. Hydrogen is thus produced and adsorbed on the catalytically acting nickel surface which subsequently converts chlorinated organic compounds into non-halogenated products, which are easily biodegradable [36].

The other reduction method for effective azo dye degradation mentioned in the literature is based on sodium borohydride ($NaBH_4$) reduction or on sodium dithionite ($Na_2S_2O_4$) produced in situ by the co-action of $NaBH_4$ and $Na_2S_2O_5$ [37].

With the emphasis on MB9, only a few articles were focused on the removal of this mentioned dye [13,18,38–41]. The previous article of our research group [13] reported the separation of MB9 using ionic liquids. As we presented in this article, the isolation method of MB9 using ionic liquids provides the effective removal of the tested dye [13]. The separation of MB9 by adsorption was consequently studied in several publications [38–40]. As shown by Singh et al., the microbial degradation of MB9 dye enables the effective discoloration of aqueous solutions contaminated with this dye. The level of MB9 HDC is low [41].

Only one article [18] focused on the oxidative degradation of MB9 using the Fenton-activated persulfate system. Pervez et al. reported the effective degradation of MB9 using the above-mentioned method expressed with discoloration and TOC removal efficiency. However, the authors did not control the possible dehalogenation of MB9 [18].

The aim of this article is to compare the effectiveness of chlorinated anionic azo dye Mordant Blue 9 (Figure 1) degradation in an aqueous solution using oxidation or reduction techniques. We compare the broadly applied Fenton reaction with innovative reduction methods.

## 2. Results and Discussion

### 2.1. Chemical Oxidation of MB9 Using $H_2O_2$

All the experiments were performed using a model aqueous solution of MB9 simulated industrial (real) wastewater from the production of acid azo dyes [42]. The applied final concentration of MB9 in reaction mixtures was 0.25 mmol/L (125 mg/L).

The initial experiments were conducted using an excess of hydrogen peroxide with overnight vigorous stirring (20 h) for the possible mineralization of MB9 dye according to Equation (1):

$$C_{16}H_9ClN_2O_8S_2Na_2 + 35\,H_2O_2 \rightarrow 16\,CO_2 + 38\,H_2O + 1\,HCl + N_2 + 2\,NaHSO_4 \quad (1)$$

In experiments of MB9 oxidation using $H_2O_2$, we demonstrated the degradation of MB9 using absorbance and discoloration efficiency measurements. The $COD_{Cr}$ was applied to determine the removal of organic compounds after oxidation quenching by the addition of 0.1 g $MnO_2$ which causes the decomposition of residual $H_2O_2$ (see the Materials and Methods section). Due to the study of chlorinated compound MB9 degradation, AOX was then determined, which expressed the content of chlorinated compounds in samples before and after degradation.

The fragmentation products of MB9 oxidation were not utilized due to complex oxidation pathways. As shown in Equation (1), theoretically at least 35 mol of $H_2O_2$ is necessary for the complete mineralization of 1 mol of MB9.

Figure 2 shows that using 20 mol of $H_2O_2$ per mol of MB9, the MB9 dye molecule was destroyed with 92% efficiency after 20 h of action according to the colour removal (measured as absorbance at absorption maximum in the visible spectrum). According to the COD determination, however, less than 10% of oxidizable organic compounds were

mineralized under these reaction conditions (using 1 mmol of $H_2O_2$ per 0.05 mmol of MB9). The gradual increase in hydrogen peroxide dosage only caused a slow increase in COD removal. Using a 40-fold molar excess of $H_2O_2$ above the MB9 dye, only around 10% of COD (and around 93% of MB9) was removed, even after 20 h of action. This observation is in apparent contradiction with the theoretical consumption of $H_2O_2$ necessary for the complete mineralization of MB9 to inorganic products (Equation (1)). Even when 200 mol of $H_2O_2$ per mol of MB9 was used, the COD value decreased to only 40% of the initial quantity (10 mmol of $H_2O_2$ used per 0.05 mmol of MB9, as shown in Figure 2).

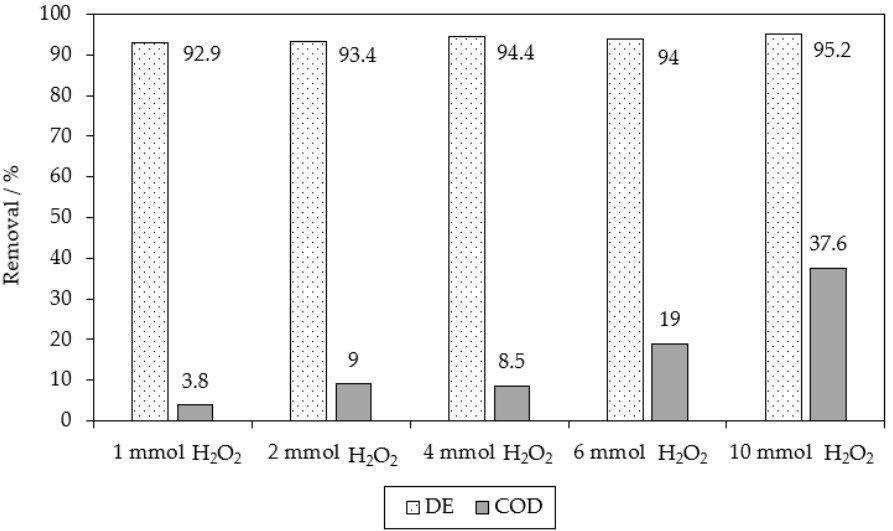

**Figure 2.** Results of the oxidative degradation of MB9 using an excess of $H_2O_2$. (Appropriate quantity of $H_2O_2$ was added to 0.05 mmol of MB9 dissolved in 200 mL of $H_2O$, with a reaction time of 20 h).

In the subsequent reaction experiments, the ferrous sulphate was added to the acidified mixture of MB9 (2 mL of 16% $H_2SO_4$ was added to 200 mL of aqueous MB9 solution containing 0.05 mmol of MB9, and the pH was adjusted in pH interval 2–2.5) and stirred with an appropriate quantity of $H_2O_2$ with the aim of utilizing the oxidation ability of OH· radical produced from the Fenton reaction (Equation (2)):

$$Fe^{2+} + H_2O_2 \rightarrow OH· + Fe^{3+} + OH^- \tag{2}$$

The applied low pH value was used with the aim of minimizing the possibility of iron (hydr)oxide precipitation during the Fenton oxidation process. As Kremer [43] reported, the optimum pH can also be lower than 3 for an efficient Fenton reaction. The studied dependence of discoloration efficiency/COD removal on the pH value is illustrated by Figure S1 in the Supplementary Materials. It was demonstrated that that there are no significant changes in discoloration efficiencies/COD removal in the range of pH values 2–4. Based on these reasons, the pH value was adjusted to pH = 2–2.5. Our experiments were therefore focused on the adjustment of the optimum dosage of $H_2SO_4$ for ensuring the acidic environment and efficient oxidation of MB9.

The addition of 2 mL of 16% $H_2SO_4$ per 200 mL of aqueous 0.25 mM MB9 solution (with a pH value of 2–2.5 and an efficient Fenton oxidation, as shown in Figure 3).

As shown in Figure 3, the Fenton oxidation initialized by the addition of $FeSO_4$ to the acidified MB9 solution significantly increased the COD removal process (compared with the action of sole $H_2O_2$). In the case of a 40-fold molar excess of $H_2O_2$ towards MB9, the ferrous sulphate addition enabled the 60% removal of oxidizable matter. By using 6 mmol of $H_2O_2$ and 0.06 mmol of $FeSO_4$ per 0.05 mmol of MB9 (molar ratio $[Fe^{2+}]/[H_2O_2]/[MB9] = 1.2:120:1$), the removal of MB9 was practically quantitative and subsequently around 82% of oxidizable matter was removed by Fenton oxidation (according to the COD removal efficiency). Similar $[Fe^{2+}]/[H_2O_2]$ ratios were reported as most

effective by Gulkaya et al. for spent dye bath oxidative treatment [19]. Without $FeSO_4$, the COD removal was low (with a COD removal efficiency below 20%).

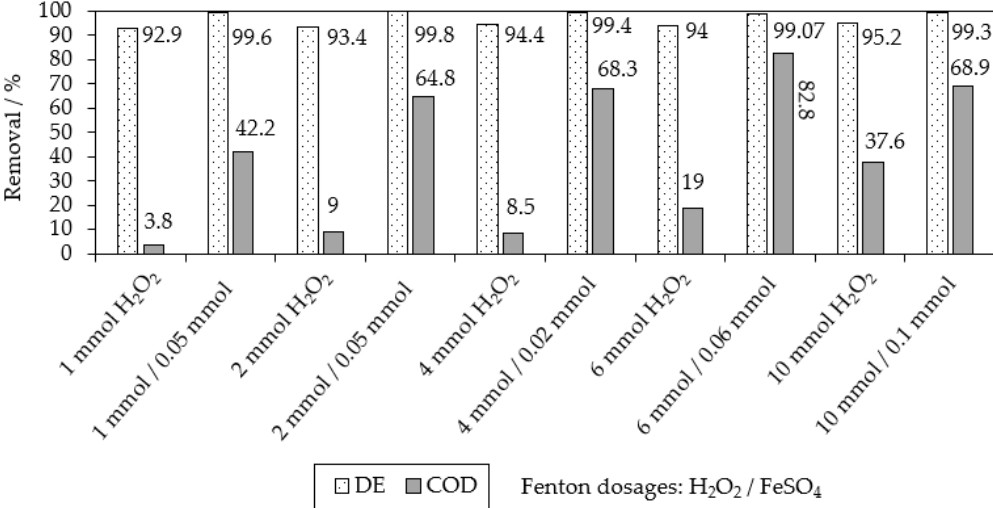

**Figure 3.** A comparison of the effect of adding $FeSO_4$ to the mixture of $H_2O_2$ and MB9 on COD and MB9 removal efficiency. ($H_2O_2$ and subsequently $FeSO_4$ were added to 0.05 mmol of MB9 dissolved in 200 mL of $H_2O$).

For a subsequent biological wastewater treatment plant (WWTP), AOX removal during the chemical oxidation of recalcitrant chemicals is most important, especially due to the biocidal effect of many chlorinated compounds such as halogenated phenols [44,45].

The MB9 removal efficiency caused by Fenton oxidation was compared with AOX removal efficiency using an excess of the Fenton reagent in an optimized molar ratio 1 mol of $FeSO_4$ per 100 mol of $H_2O_2$. As is illustrated in Figure 4, both AOXs and decolorization efficiencies were quite similar (88% of MB9 removal and 86% of AOX removal) and high enough using the molar ratio $[Fe^{2+}]/[H_2O_2]/[MB9] = 0.2:20:1$, even after 4 h of action.

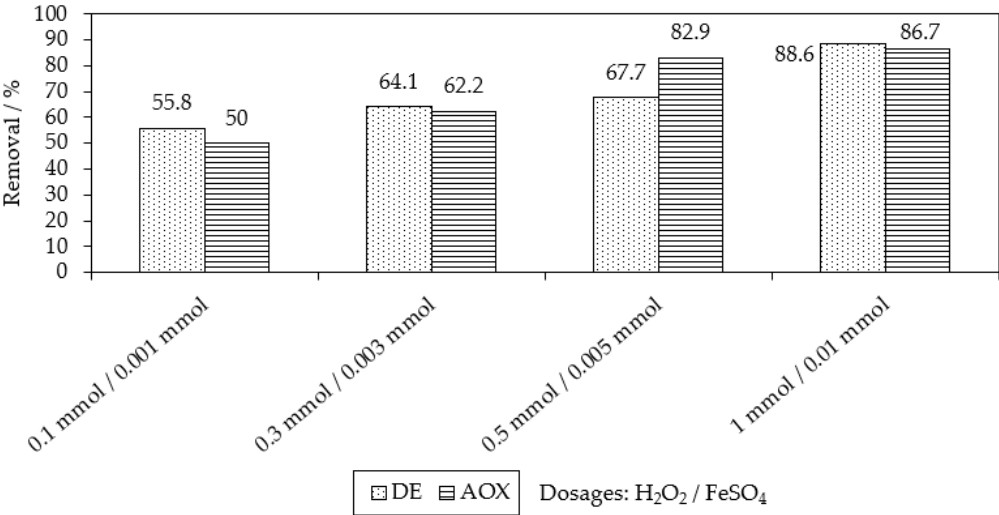

**Figure 4.** A comparison of MB9 oxidation using the Fenton reaction after 4 h of action. ($H_2O_2$ and subsequently $FeSO_4$ were added to 0.05 mmol of MB9 dissolved in 200 mL of $H_2O$).

We observed that the prolonged reaction time of Fenton oxidation is an important parameter for the maximization of AOX and COD removal (Figure 5 below). Due to this reason reaction time, 20 h was applied in the subsequent experiments.

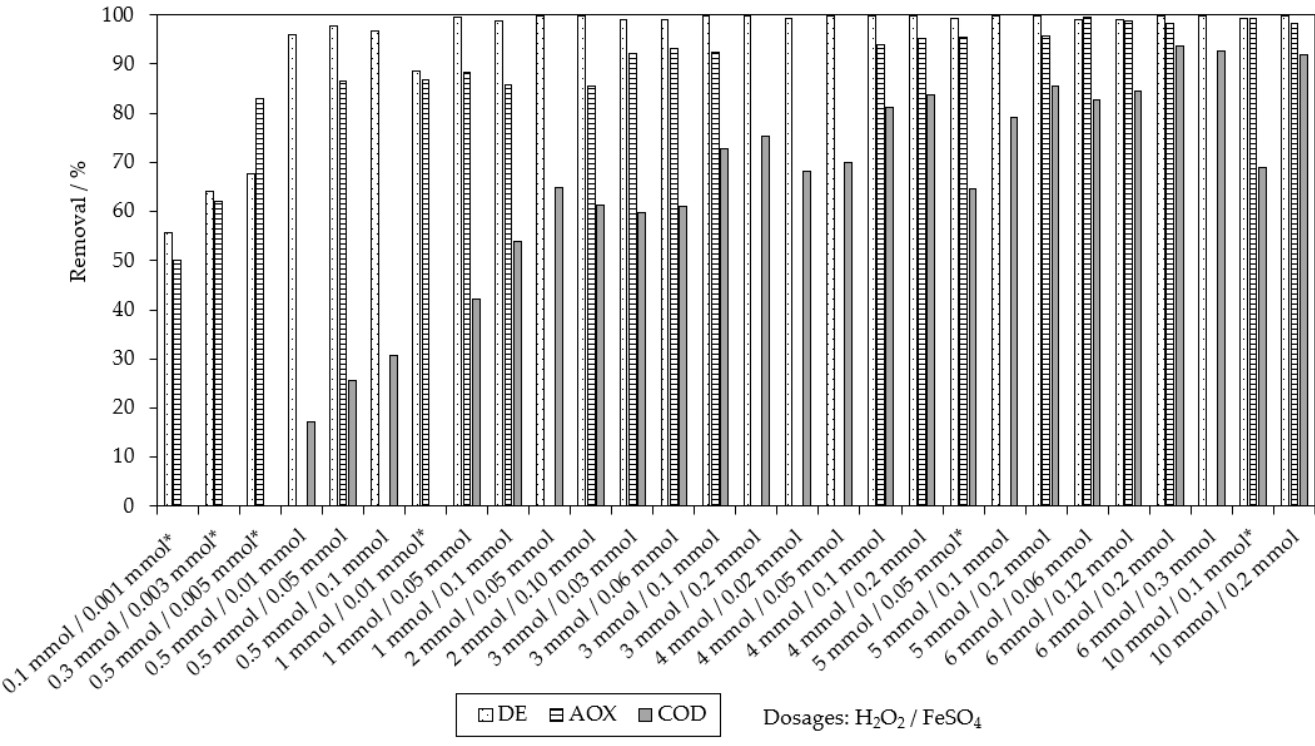

**Figure 5.** Summary of removal efficiencies obtained using different molar ratios between $[Fe^{2+}]/[H_2O_2]/[MB9]$. (* $H_2O_2$ and subsequently $FeSO_4$ were added to acidified 0.05 mmol of MB9 in 200 mL of $H_2O$, with a reaction time of 20 h or 4 h).

The effect of different molar ratios $[Fe^{2+}]/[H_2O_2]/[MB9] = 1–4{:}10–120{:}1$ is illustrated in Figures 5 and 6. Comparing the relationship between the molar ratios of $FeSO_4$ and $H_2O_2$, $[Fe^{2+}]/[H_2O_2] = 1{:}20–30$ seems to be more effective. It is evident that for the fairly complete removal of MB9 and AOX, at least 120 mol of $H_2O_2$, together with 4 mol of $FeSO_4$ per mol of MB9, should be used (Figures 6 and 7). This means that the actual minimal excess of $H_2O_2$ is more than 3–3.4 times higher compared to the theoretical consumption of $H_2O_2$ for the complete mineralization of MB9 (see Equation (1)).

Despite the high discoloration efficiency of MB9 using Fenton reaction after 4 h (see Figure 7), the removal of COD and AOX did not amount to satisfactory results, as shown in Figure 6. Therefore, the prolonged reaction time (20 h) in combination with high dosages of $H_2O_2/Fe^{2+}$ was proven to be necessary for the effective degradation of MB9 (including AOX and COD removal).

Under described conditions, however, the $[Fe^{2+}]/[H_2O_2]$ ratio seems to be far from optimal [46], likely accompanied by a high consumption of $H_2O_2$. The real excess of $H_2O_2$ was more than 3–3.5-times higher compared to the theoretical consumption (see Equation (1)).

Due to the reasons mentioned above, optimization experiments were performed with the aim of rationalizing the dosage of components of the Fenton reagent.

Compared with the persulfate-based method [18], the required excess of $H_2O_2$ for the practically complete removal of AOX and COD seems to be high. However, the published persulfate method only determined residual TOC and MB9 removal, not a concentration of hardly biodegradable and toxic AOXs. For 95% discoloration, a similar 10-fold molar excess of $H_2O_2$ and an equimolar quantity of $FeSO_4$ toward MB9 should be used (Figure 5, second column from the left).

Finally, it is evident that the subsequent neutralization of the reaction mixture after Fenton oxidation increases the additional removal of MB9 (almost a 20% increase in the COD removal efficiency) caused by the coagulation of iron (hydr)oxides, as shown in the Supplementary Materials, Table S1. By neutralizing the acidic reaction mixture, in

contrast, the produced solid by-product (iron (hydr)oxides containing slurry) is potentially contaminated with adsorbed oxidation (chlorinated) products. This slurry is categorized as a dangerous waste and must be disposed in a safe manner [47].

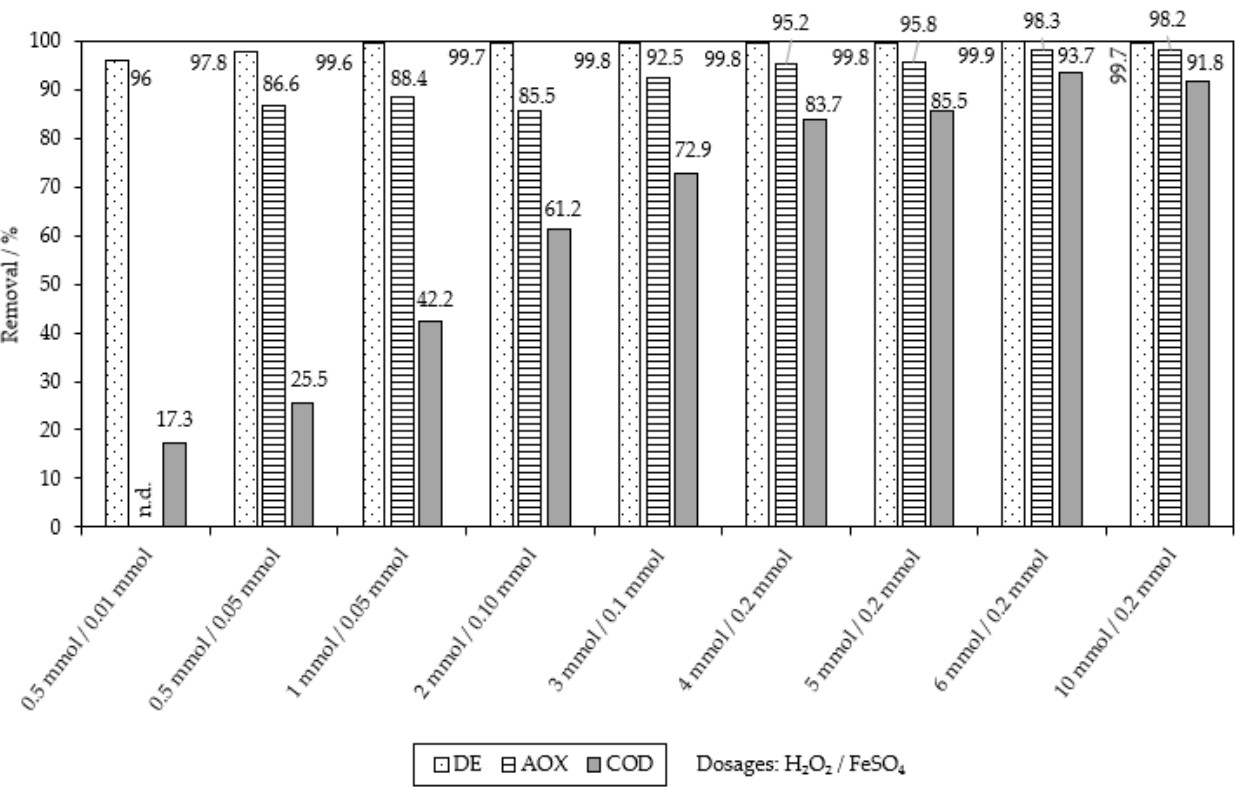

**Figure 6.** The effect of different $FeSO_4$ and/or $H_2O_2$ dosages on the removal efficiencies. ($H_2O_2$ and subsequently $FeSO_4$ were added to acidified 0.05 mmol of MB9 dissolved in 200 mL of $H_2O$, with a reaction time of 20 h).

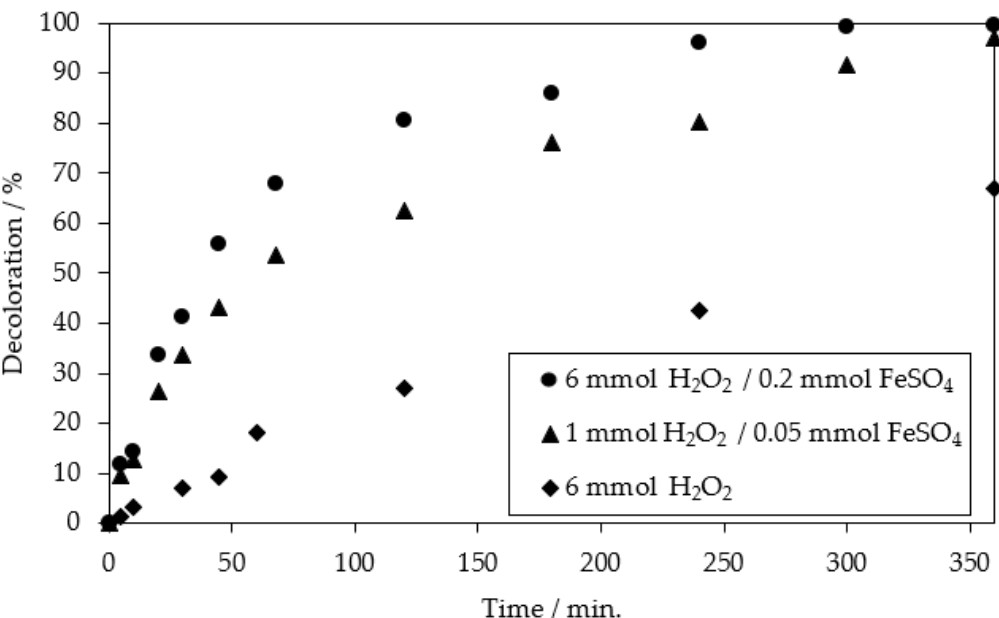

**Figure 7.** A comparison of the dependence of MB9 oxidative degradation on time under different dosages of $H_2O_2$ and $FeSO_4$. (Appropriate quantities of $H_2O_2$ and $FeSO_4$ were added to 0.05 mmol of MB9 dissolved in 200 mL of $H_2O$).

### 2.2. The Chemical Reduction of MB9

Compared with chemical oxidation, the chemical reduction of pollutants was described as more selective [30,31]. A higher selectivity is joined with a lower consumption of reagents.

In the first set of experiments, iron turnings (ZVI) were used for the reductive degradation of MB9.

The obtained results are illustrated in Figure 8. ZVI is a strong reducing agent, and moreover ZVI is easily produced and inexpensive [32]. It has already been rendered to be effective in the reductive degradation of selected chlorinated organic derivatives [48]. In some studies [33,34], methods of dye reduction using micro or nano iron have been described.

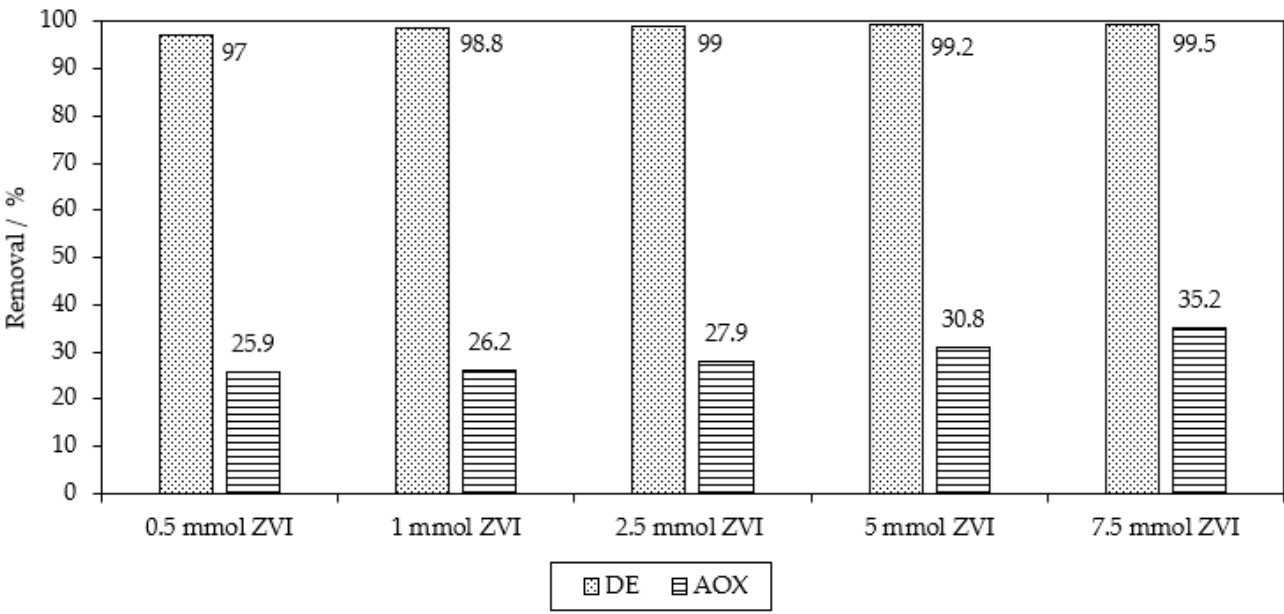

**Figure 8.** The reductive degradation of MB9 using ZVI. (ZVI was added to aqueous solution containing 0.05 mmol of MB9 in 100 mL of $H_2O$ acidified with 40 mmol of $H_2SO_4$, with a reaction time of 3 h).

Even using a 10-fold molar excess of ZVI, 97% of MB9 was removed. The AOX removal efficiency, however, was low (25.9%). This corresponds with the known fact that ZVI is not a very effective dechlorination agent in the case of chlorinated aromatic compounds. In the case of a 150-fold molar excess of ZVI, only a 35.2% AOX removal efficiency was observed (together with a 99.5% MB9 removal efficiency).

Apart from ZVI, metallic aluminum (granules, powder, or foil) was tested for the reductive degradation of MB9 dye. Aluminum was reported to be more effective in wastewater treatment compared with ZVI [35]. Figure 9 illustrates differences in $Al^0$-based reduction efficiencies using diluted sulfuric acid. In addition to $Al^0$, the Raney Al-Ni alloy, known to be very effective reductant, was used [30,31]. The obtained degradation efficiencies were slightly worse using $Al^0$ compared with $Fe^0$.

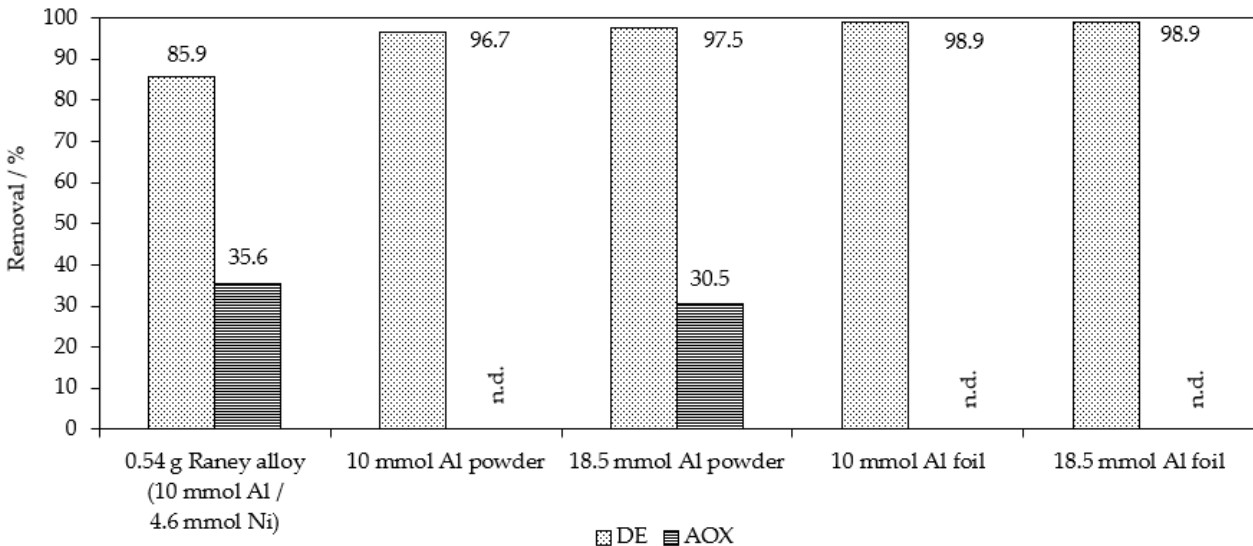

**Figure 9.** The reductive degradation of MB9 using different types of $Al^0$ in an acidic environment. ($Al^0$ was added to the solution of 0.05 mmol of MB9 and 40 mmol of $H_2SO_4$ in 100 mL of $H_2O$, with a reaction time of 20 h) (n.d. = not determined).

It is well known that $Al^0$ is a powerful reductant, especially in diluted alkali metal hydroxide solutions [35]. For this reason, alkaline conditions were used for subsequent experiments. The obtained results are presented in Figure 10.

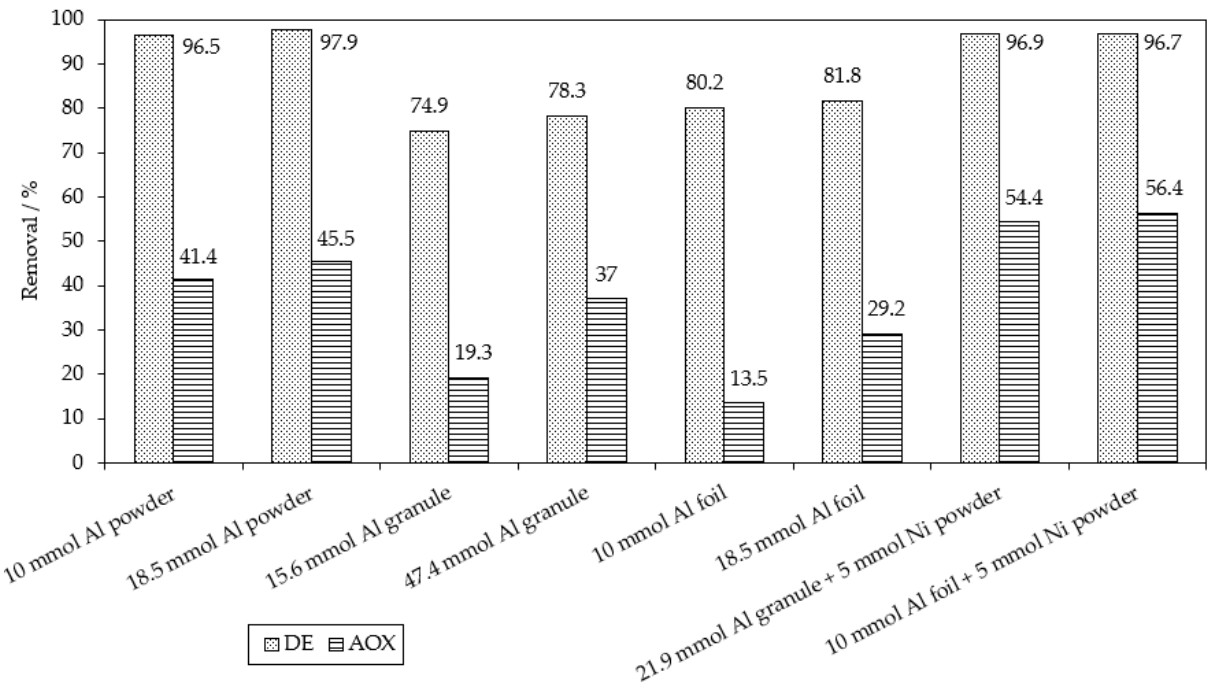

**Figure 10.** The reductive degradation of MB9 using different types of $Al^0$ and their mixtures with nickel in an aqueous 50 mM NaOH solution. ($Al^0$ and $Ni^0$ were added to the solution of 0.05 mmol of MB9 and 5 mmol of NaOH in 100 mL of $H_2O$, with a reaction time of 3 h).

When using Al powder in particular, both MB9 and AOX removals were higher compared with $ZVI/H_2SO_4$ or $Al^0/H_2SO_4$ actions. In addition, the used Al granules were similarly active as Al foil, although both above-mentioned forms of Al were less active compared with Al powder. In the case of the used Al granules, a low specific surface seems

to be the reason for the low reduction activity. In addition, the kinetic comparison indicates that the reaction rate of $Al^0$/NaOH was higher (Figure 11).

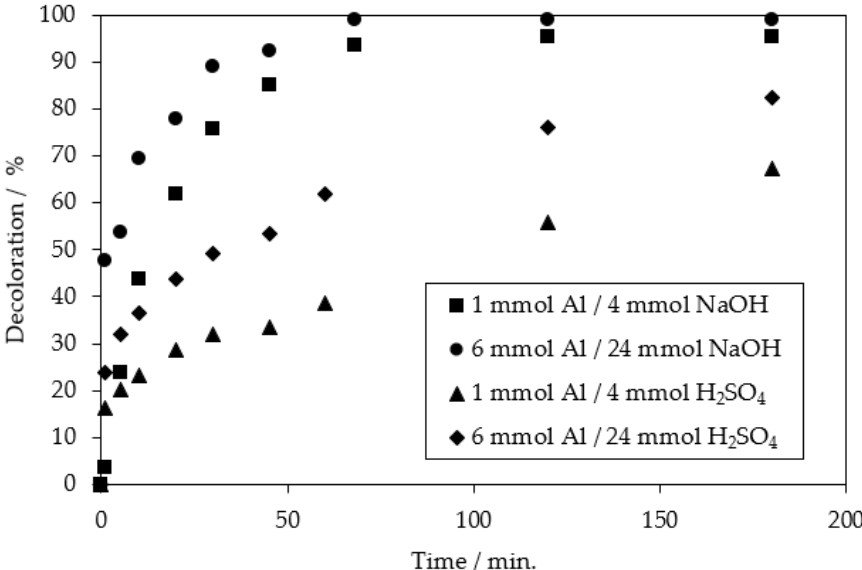

**Figure 11.** A comparison of rates of MB9 reductive degradation under different dosages of Al and NaOH or $H_2SO_4$. (Appropriate quantities of Al and NaOH/$H_2SO_4$ were added to 0.05 mmol of MB9 dissolved in 200 mL of $H_2O$).

Due to our excellent experience with the hydrodehalogenation activity of Al-Ni alloys [30,31], we tried to use a mixture of Ni powder together with $Al^0$ (Figure 10) and Raney Al-Ni alloys for MB9 reductive treatment (Figure 12). After the addition of Ni powder, both the MB9 and AOX removal efficiencies increased in comparison with the action of $Al^0$ alone. Using the Raney Al-Ni alloy, however, the HDC of MB9 reduction products was much higher compared with the action of a mixture of $Al^0$ and $Ni^0$ powders.

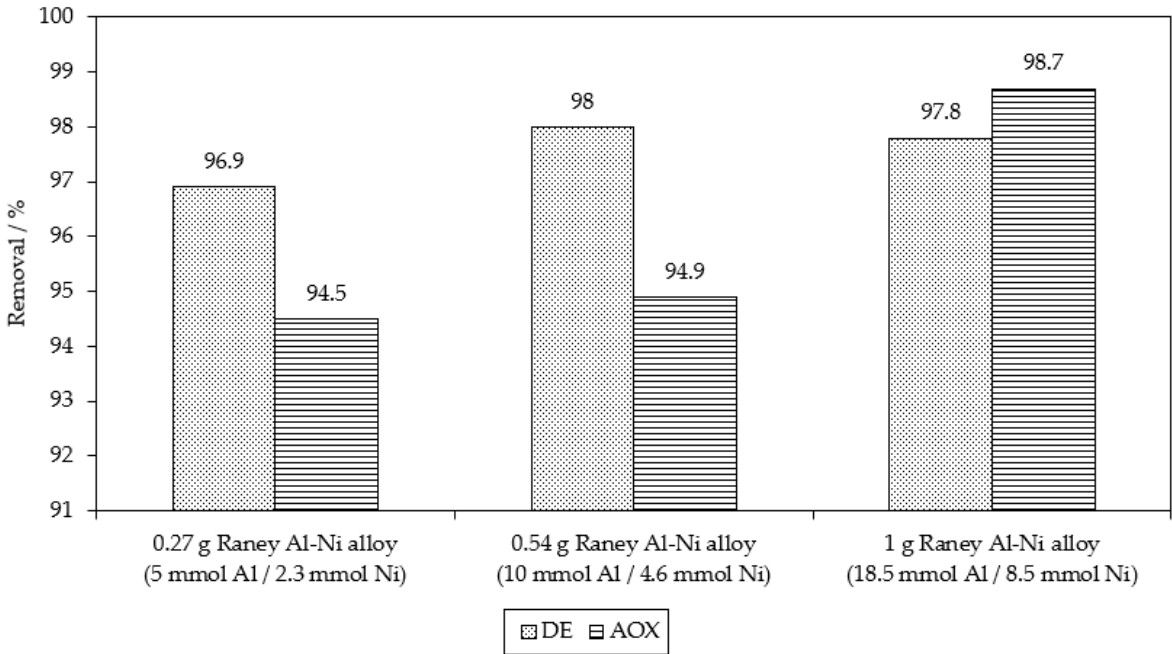

**Figure 12.** A comparison of AOX and MB9 removal caused by Raney Al-Ni alloy action. (Al-Ni alloy added to aqueous solutions of 0.05 mmol of MB9 and 5 mmol of NaOH in 100 mL of $H_2O$, with a reaction time of 3 h).

In addition, by using the Raney Al-Ni alloy instead of $Al^0 + Ni^0$ powders, the removal of AOX (hydrodechlorination, HDC) was very effective (Figure 12) [30,31]. HDC was completed after 3 h of Al-Ni alloy/NaOH action. In comparison with the Fenton reaction, HDC using Al-Ni alloy/NaOH processed almost seven times faster in the degradation of MB9. The proposed reaction pathway for the reductive treatment of MB9 using $Al^0 + Ni^0$ or Al/Ni alloys is depicted in Scheme 1.

**Scheme 1.** Proposed reaction pathway for the reduction of MB9 accompanied by hydrodechlorination.

A combination of the rapid azo bond reduction of the MB9 structure using cheaper $Al^0$ powder (pretreatment) and subsequently HDC using the Al-Ni alloy (see Scheme 1) serves as a simple cost-effective application in the degradation of non-biodegradable chlorinated dye in aqueous streams. Non-chlorinated products based on the reduction/HDC of this promising reductive method were proved using the LC-MS method. As indicated by LC-MS analysis (see Supplementary Material, Figure S2), the HDC of MB9 using Al-Ni (Figure 12, first column from the left, 3 h of action) proceeded smoothly according to the HDC reaction depicted in Scheme 1.

The starting Raney Al-Ni alloy and its reaction products were studied by X-ray powder diffraction in our earlier work [30]. The changes in surface morphology were also documented by SEM [30]. As we published earlier, the starting Raney Al-Ni alloy contained 41% $Ni_2Al_3$, 57% $NiAl_3$, and 2% aluminum. Briefly, after the beginning of the reaction (after 30 min of HDC of chloroaromates (using 0.27 g of Al-Ni/1 mmol Ar-Cl + 50 mmol of NaOH), the original Raney Al-Ni alloy changed slightly, the pure aluminum disappeared, and the $NiAl_3$ alloy quantity decreased (34% $NiAl_3$ and 66% $Ni_2Al_3$). After 60 min of the reaction, the $NiAl_3$ diffraction lines disappeared completely, and the isolated solid part only contained $Ni_2Al_3$. At the end of the HDC reaction, apart from the rest of $Ni_2Al_3$ (18%), the isolated solid mainly contained $Ni_{0.92}Al_{0.08}$ alloy (61%) and pure metallic nickel (10%), together with a low quantity of bunsenite (2% NiO) [30].

In addition, after the HDC, the produced nickel sludge was simply removable by sedimentation and/or filtration. The deactivated Ni catalyst can be regenerated and recycled with several benefits [49].

The dissolved aluminum in the alkaline reaction mixture after HDC can be utilized for the subsequent sorption of HDC products via the coagulation and flocculation of $Al(OH)_3$ produced by alkaline aqueous phase neutralization after the HDC procedure [30,31,49–51].

The above-mentioned coagulation also increased the COD removal efficiency by up to 15% (see the Supplementary Materials, Table S1).

The HDC of MB9 using Al-Ni therefore effectively removed AOX, and, subsequently, the neutralization of the reaction mixture, accompanied by coagulation, facilitated additional COD removal. In contrast with the Fenton reaction, this slurry after coagulation

contains more biodegradable (non-chlorinated) HDC products and it can be used as an additive for cement production.

In the above-mentioned Al-based MB9 removal methods, the sorption of MB9 on the surface of Al-based reductive agents could play an important role. The sorption experiments were performed to compare the sorption effect on MB9 removal using different sources of $Al^0$.

A comparison of sorption efficiencies is illustrated in Figure 13 (the contact time was 3 h). The highest removal efficiency of MB9 was obtained with $Al^0$ powder (MB9 65.8%) or $Al^0$ foil (59.5%). A similar removal efficiency was achieved by the Raney Al-Ni alloy (60%), as shown in Figure 13. $Al^0$ granules were not very efficient for MB9 sorption, possibly due to the low specific surface area of Al granules.

In addition, the calculated sorption parameter—adsorption capacity $q$ (mg/kg)—demonstrated that the sorption capacities of Al powder and Al foil for MB9 dye were 65.8 mg/kg and 59.5 mg/kg, respectively. In contrast, the adsorption capacity of the Raney Al-Ni alloy was only 30 mg/kg.

Despite this fact, the results indicate that MB9 is selectively adsorbed, especially on powders of aluminum or Raney Al-Ni alloys or Al foil. From the above-mentioned results, it can be deduced that the main drawback of the used Al foil in the reduction process is its very rapid dissolution (consumption) under the used reaction conditions (over a few minutes, as was observed), which prevents the effective MB9 reductive removal.

A comparison of decolorization rates based on the application of $Al^0$ in alkaline and acidic reaction conditions is illustrated in Figure 14. It is evident that the removal of MB9 dye is the most rapid when the $Al^0$-based reduction process is used in aqueous NaOH. The reaction rates of the Fenton oxidation and $Al^0$-based reduction of aqueous $H_2SO_4$ were quite similar.

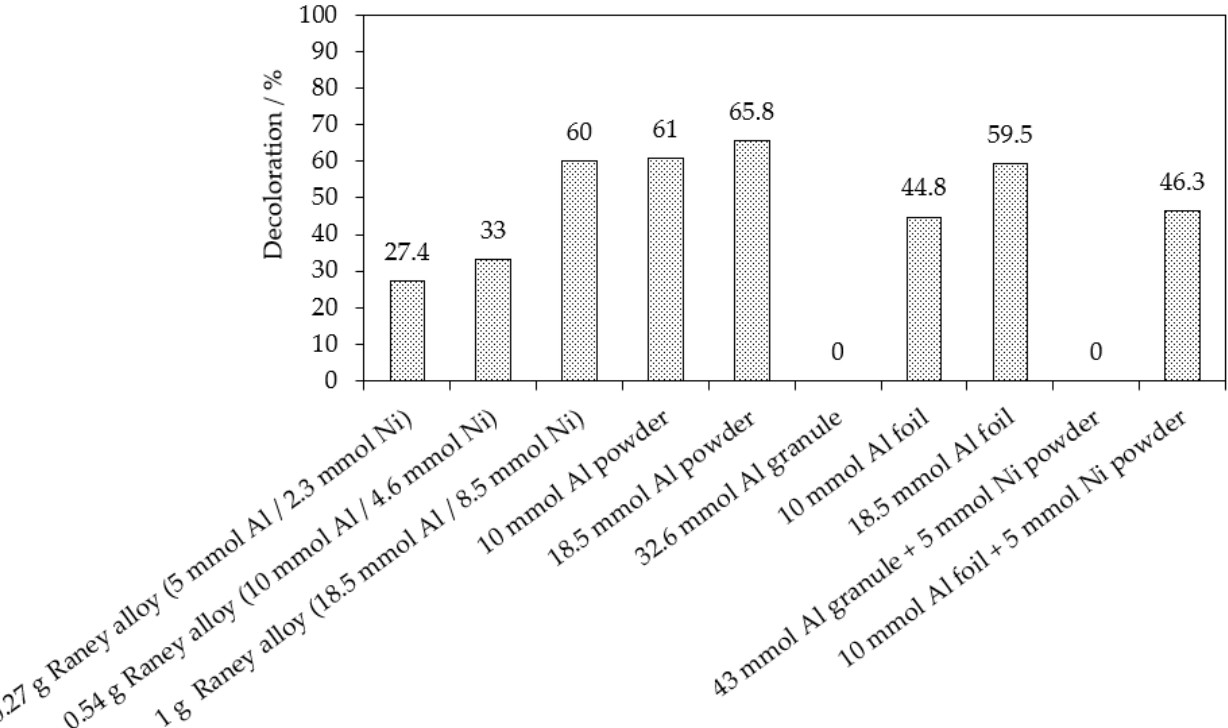

**Figure 13.** A comparison of sorption abilities of tested Al-based reduction agents. (Mentioned quantities of metal(s) were added to the 0.05 mmol of MB9 dissolved in 100 mL of $H_2O$, with a sorption time of 3 h).

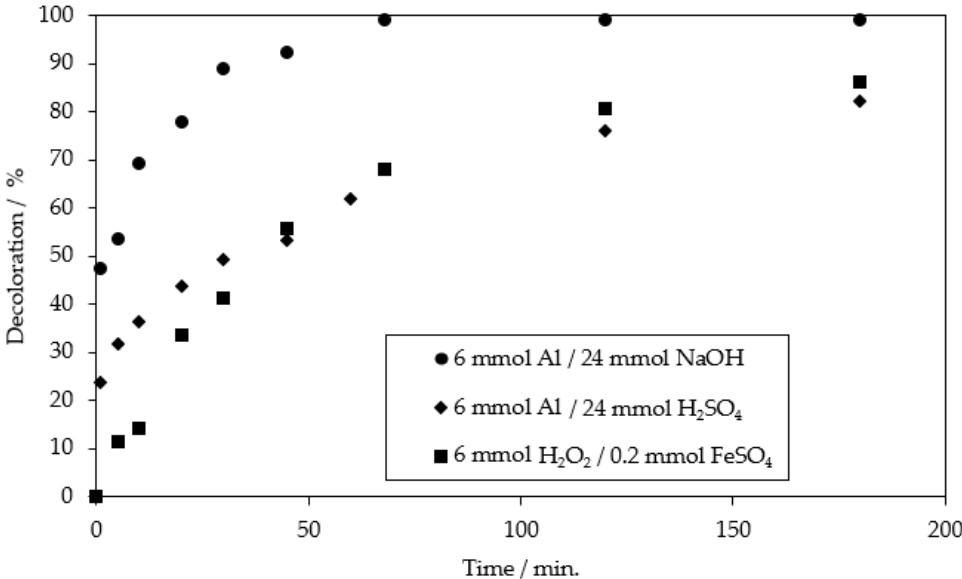

**Figure 14.** A comparison of discoloration of MB9 using of 6 mmol of oxidant/reductant. (An appropriate quantity of reagents was added to 0.05 mmol of MB9 dissolved in 200 mL of $H_2O$).

Another reduction method for effective azo dye degradation mentioned in the literature is based on sodium borohydride ($NaBH_4$) or sodium dithionite ($Na_2S_2O_4$, see reactions 3 and 4) produced in situ by the reaction of $NaBH_4$ with $Na_2S_2O_5$ [37] (Scheme 2). It was verified that sodium borohydride caused efficient MB9 reduction when applied alone; however, the AOX removal was sluggish (Figure 15).

$$S_2O_5^{2-} + H_2O \rightarrow 2\ HSO_3^- \tag{3}$$

$$BH_4^- + 8\ HSO_3^- + H^+ \rightarrow 4\ S_2O_4^{2-} + B(OH)_3 + 5\ H_2O \tag{4}$$

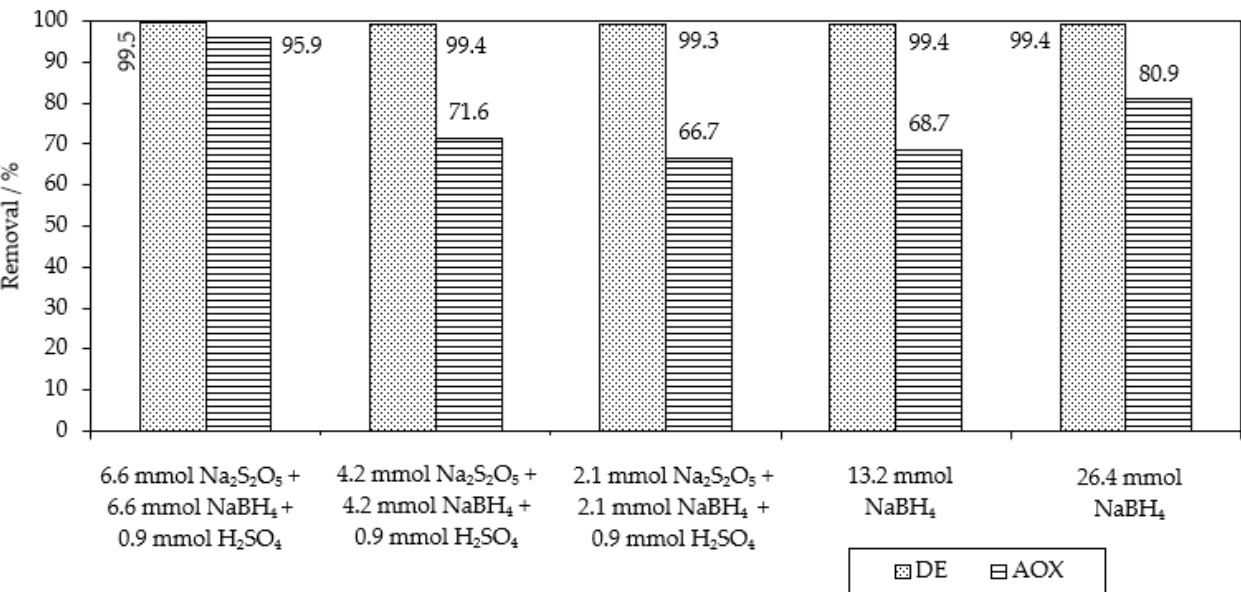

**Figure 15.** Reductive removal of MB9 and AOX using reductant(s) based on $NaBH_4$ action. (The above-mentioned reduction agents were added to the aq. solution of 0.1 mmol of MB9 in 200 mL of $H_2O$, with a reaction time of 20 h).

**Scheme 2.** The proposed reaction scheme of MB9 reductive degradation using $NaBH_4/Na_2S_2O_5$.

When applying a combination of cheap $Na_2S_2O_5$ accompanied by the subsequent addition of $NaBH_4$, both MB9 and AOX were removed very effectively (Figure 15). The proposed reaction scheme of MB9 reductive degradation using $NaBH_4/Na_2S_2O_5$ is illustrated in Scheme 2. As indicated by LC-MS analysis (see Supplementary Material, Figure S3), a smooth reduction of MB9 is depicted in Scheme 2.

Table 1 shows a comparison of the above-described results of the oxidation and reduction of MB9 (calculated per one mol of tested dye for the demonstration of practical application). The reaction rates of the tested reduction processes are compared in Figure 16. It is evident that the homogeneous reduction using $NaBH_4/Na_2S_2O_5$ is the most rapid process weighted by decolorization. A comparison of reagents consumption and degradation efficiencies in performed optimized experiments is depicted in Figure 17. The pretreatment of MB9 solutions using $Al^0/NaOH$ and subsequently HDC with Al-Ni/NaOH also facilitates the effective and smooth removal of MB9 dye.

In summary, the Fenton reaction is cheaper than other tested reductive systems (see Supplementary Materials, Table S2) due to the low cost of hydrogen peroxide. The reaction rate, however, is slow, even when using an excess of $H_2O_2$ that is more than 3.5 times higher than that using the theoretical consumption. On the other hand, for the practical rapid removal of AOX necessary for scale-up, reductive systems such as $NaBH_4/Na_2S_2O_5$ or a combination of $Al^0/Al$-Ni alloys are more utilizable.

The above-mentioned combination of $NaBH_4/Na_2S_2O_5$ or $Al^0/Al$-Ni alloys provides an increase in the removal of MB9 from model wastewater. These methods also offer a decrease in price compared with less efficient/more expensive reductions using $NaBH_4$ or Al-Ni alloy alone (see the Supplementary Materials, Table S2).

Moreover, HDC using Al-Ni alloys proceeds smoothly in a few hours in contrast with HDC using the Fenton reaction (reaction time for effective MB9 removal 20 h, as shown in Table 1). The lower reaction time of the proposed reductive process saves energy and further decreases economic costs. In addition, the application of the Al-Ni alloy enables the recycling of produced Ni-slurry [49–51]. This recycling of used Al-Ni alloys also potentially decreases the price of this effective HDC method.

**Table 1.** A comparison of MB9 removal efficiencies caused by oxidation or reduction using different reactants at room temperature (calculated per mol of MB9).

| Applied Degradation Method | Quantity of Used Reagents per mol of MB9 | Efficiency of MB9 Discoloration | Other Removal Efficiencies (%) | Ref. |
|---|---|---|---|---|
| Persulfate-based AOP at pH = 4.88 | 16 mol of $Na_2S_2O_8$ + 14 mol of $FeSO_4$ | 97% (after 30 min) | 60% of TOC (after 30 min) | [18] |
| Fenton oxidation at pH = 4.88 | 16 mol of $H_2O_2$ + 14 mol of $FeSO_4$ | 48% of MB9 (after 30 min) | 38% of TOC (after 30 min) | [18] |

**Table 1.** *Cont.*

| Applied Degradation Method | Quantity of Used Reagents per mol of MB9 | Efficiency of MB9 Discoloration | Other Removal Efficiencies (%) | Ref. |
|---|---|---|---|---|
| Fenton oxidation at pH = 2–2.5 | 28 mol of $H_2SO_4$ 120 mol of $H_2O_2$ + 4 mol of $FeSO_4$ | 86.1% of MB9 (after 3 h) | 37.6% of COD (after 3 h) | This work |
| Fenton oxidation at pH = 2–2.5 | 28 mol of $H_2SO_4$ 120 mol of $H_2O_2$ + 4 mol of $FeSO_4$ | >99% of MB9 (after 20 h) | 98.3% of AOX and 93.7% of COD (after 20 h) | This work |
| Chemical reduction ZVI | 150 mol of $Fe^0$ 800 mol of $H_2SO_4$ | 99.5% of MB9 (after 3 h) | 35.2% of AOX (after 3 h) | This work |
| Chemical reduction $Al^0$ powder | 370 mol of $Al^0$ 800 mol of $H_2SO_4$ | 97.5% of MB9 (after 20 h) | 30.5% of AOX (after 20 h) | This work |
| Chemical reduction $Al^0$ powder | 200 mol of $Al^0$ 100 mol of NaOH | 96.5% of MB9 (after 3 h) | 41.4% of AOX (after 3 h) | This work |
| Chemical reduction $Al^0$ powder + $Ni^0$ powder | 200 mol of $Al^0$ 50 mol of $Ni^0$ 100 mol of NaOH | 96.7% of MB9 | 56.4% of AOX (after 3 h) | This work |
| Chemical reduction Raney Al-Ni | 100 mol of $Al^0$ used in Al-Ni 100 mol of NaOH | 96.9% of MB9 (after 3 h) | 94.5% of AOX (after 3 h) | This work |
| Chemical reduction $Al^0$ powder + Raney Al-Ni | 120 mol of $Al^0$ 480 mol of NaOH 44 mol of $Al^0$ (in Al-Ni) 480 mol of NaOH | 99% of MB9 (after 3 + 20 h) | 91.7% of AOX (after 3 + 20 h) | This work |
| Chemical reduction $NaBH_4$ | 132 mol of $NaBH_4$ | 99.4% of MB9 (after 20 h) | 68.7% of AOX (after 20 h) | This work |
| Chemical reduction $NaBH_4/Na_2S_2O_5$ | 66 mol of $NaBH_4$ + 66 mol of $Na_2S_2O_5$ + 18 mol of $H_2SO_4$ | 99.5% of MB9 (after 30 min) | 95.9% of AOX (after 20 h) | This work |

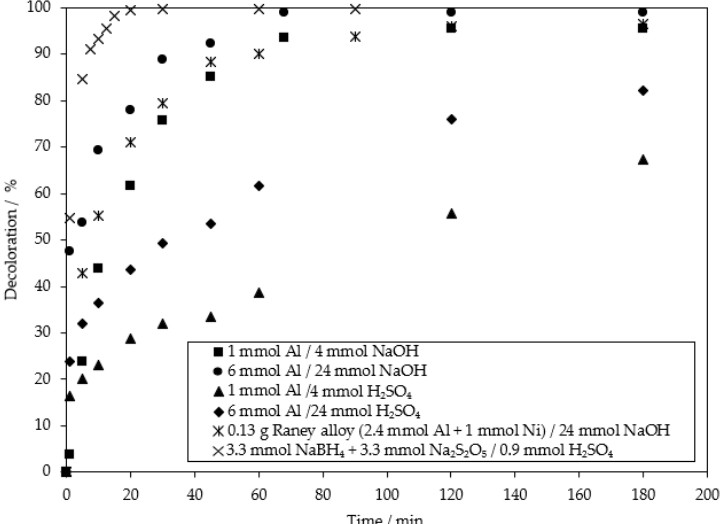

**Figure 16.** A comparison of the reaction rates of MB9 reductive degradation under different dosages of reductants. (An appropriate quantity of reagents was added to 0.05 mmol of MB9 dissolved in 200 mL of $H_2O$).

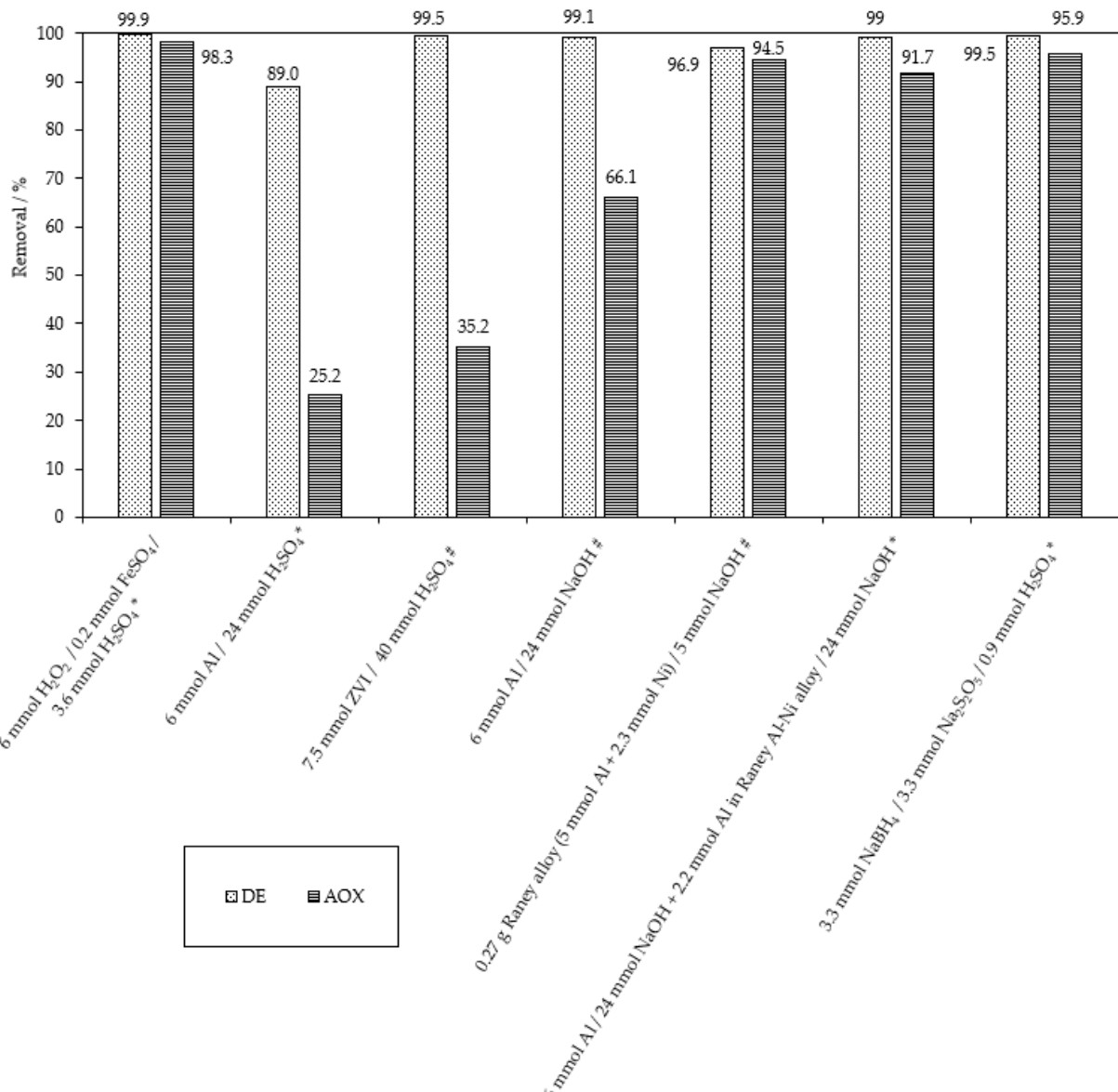

**Figure 17.** A comparison of reagent consumption and effectiveness of MB9 degradation. (An appropriate quantity of reagents was added to 0.05 mmol of MB9 dissolved in 200 mL of $H_2O$, with a reaction time of # 3 h or * 20–21 h).

## 3. Materials and Methods

### 3.1. Reagents and Materials

The acidic azo dye Mordant Blue 9 (MB9) was purchased from commercial sources in a defined purity 50% of MB9 (Sigma Aldrich Co., Prague, Czech Republic). Sigma-Aldrich does not provide the exact composition of the other ingredients. It can be assumed, however, that other ingredients are inorganic salts from dye manufacturing [6].

$H_2O_2$ (30%) and $FeSO_4.7H_2O$ (p.a. quality) were obtained from Lach-Ner Co., Neratovice, Czech Republic. The Raney Al-Ni alloy, $NaBH_4$, and $Na_2S_2O_5$ were supplied from Sigma Aldrich Co., Prague, Czech Republic. The powdered $Al^0$ (Labo MS Co., Prague, Czech Republic), granulated Al (Lachema Co., Brno, Czech Republic), Al foil, powdered Ni (Alfa Aesar, Heysham, Lancashire, UK), and zero-valent $Fe^0$ (ZVI, local supplier Synthesia Co., Pardubice, Czech Republic) were purchased in purity higher than 95%. Additional chemicals ($H_2SO_4$, NaOH and $MnO_2$) in p.a. quality were obtained from Lach-Ner Co.,

Neratovice, Czech Republic. Finally, all the aqueous solutions were prepared from demineralized water.

### 3.2. Oxidative and Reductive Degradation of MB9 Procedures

The comparative degradation experiments were performed in a 250 mL round-bottomed flasks equipped with magnetic stirring on Starfish equipment (Radleys Discovery Technologies, Saffron Walden, UK), installed on a magnetic stirrer Heidolph Heistandard for parallel reactions. The reaction flasks were closed by a tube filled with granulated charcoal. After the appropriate time period of stirring, the reaction mixtures were filtered and analyzed.

In the case of the oxidative degradation of MB9 using hydrogen peroxide, all the samples were mixed before filtration with 0.1 g of $MnO_2$ used as the catalyst for the rapid decomposition of unreacted $H_2O_2$. The final measurement of residual $H_2O_2$ in the reaction mixtures was not required and the interferences of COD determination were negligible (unreacted $H_2O_2$ significantly increased the COD value, as verified). For the Fenton reaction, the pH values of reaction mixtures were modified using the addition of 10 mL of 16% $H_2SO_4$ per liter of 0.5 mM aqueous MB9 solution (2 mL of 16% $H_2SO_4$ per 200 mL of $H_2O$ containing 0.05 mmol of MB9). The pH values were measured and/or controlled by a contact pH meter Orio Star 10 (Thermo Fisher Scientific, Pardubice, Czech Republic).

In the case of the ZVI application, due to the limitation of the magnetic stirring influence, the reaction mixtures were intensively stirred. Thus, the sufficient contact of ZVI with the aqueous solutions of MB9 was ensured.

### 3.3. Chemical Analysis

A Hach DR2800 (Austria) VIS spectrophotometer was employed for the absorbance measurements using 1 cm glass cuvettes. The concentrations of MB9 were determined by measuring at a wavelength 516 nm. Analyses of the COD were carried out in accordance with the ISO EN 9562 standard and aqueous solutions were determined using the Hach Lange cuvette test using the Hach DR2800 (Austria) VIS spectrometer. The AOXs were analyzed according to the European ISO 9562 standard using the Multi X 2500 analyzer (Analytic Jena GmbH., Jena, Germany).

The discoloration of MB9 solutions was calculated according to Equation (5):

$$DE = \left(1 - \frac{A}{A_0}\right) \times 100 \tag{5}$$

where *DE* is the discoloration efficiency of MB9 aqueous solution (%), *A* is the measured absorbance after the degradation process, and $A_0$ is the initial absorbance of MB9 solution.

The removal efficiency (%) of COD or AOX from the model aqueous solutions of MB9 was evaluated by Equation (6):

$$RE = \left(1 - \frac{c}{c_0}\right) \times 100 \tag{6}$$

where *RE* is removal efficiency of AOX or COD (%), *c* is the concentration of AOX or COD in the solution after the degradation process (mg/L), and $c_0$ is the initial concentration of AOX or COD in the MB9 solution before the degradation process (mg/L).

The reduction products after the reductive degradation of MB9 were detected using the LC-MS technique with additional UV detection. The water samples were properly diluted and 2 μL of the diluted sample was injected in the Hypersil Gold C4 column (100 × 3 mm). The LC separation of the detected compounds was performed in a LC-MSD TRAP XCT Plus system (Agilent Technologies, Santa Clara, CA, USA) and was carried out with a gradient program using (A) 5 mM $CH_2COONH_2$, (B) 95% $CH_3COONH_4$, and (C) acetonitrile at a flow rate of 0.5 mL/min and at temperature of 40 °C. The detected compounds were completely separated under the established gradient program in 40 min. The UV detection was carried out at a wavelength 254 nm. The LC system was connected to

an Agilent 6490 Triple Q MS with an electrospray interface (ESI) and was operated in both positive and negative ionization mode, using the above-described gradient program in both cases. The MS settings were as follows: a drying gas temperature of 350 °C; a drying gas of 10 L/min; ion source gas of 50 psi; and a mass range of 50–1200 Da. Nitrogen was used as the nebulizer gas, curtain gas, and collision gas.

## 4. Conclusions

This paper aims to compare the efficiency of common Fenton oxidation together with innovative chemical reduction of chlorinated anionic azo dye Mordant Blue 9 to simply biodegradable nonchlorinated products.

The classic Fenton oxidation is broadly used and is a frequently evaluated treatment method using cheap and low toxic reagents, such as hydrogen peroxide and ferrous sulphate. These reagents cause the negligible secondary contamination of treated and subsequently neutralized water free of insoluble impurities such as iron (hydr)oxides.

We demonstrated that an appropriate excess of hydrogen peroxide (120 mol), $H_2SO_4$ (28 mol), and $FeSO_4$ (4 mol) is able to destroy chlorinated acid azo dye Mordant Blue 9 to non-chlorinated by-products with the subsequent removal of oxidizable organic compounds after 20 h of action. By neutralizing the acidic reaction mixture, the produced solid by-product, including iron (hydr)oxides containing slurry, is potentially contaminated with adsorbed oxidation products. This slurry is categorized as dangerous waste and must be disposed in a safe manner. In addition, the large consumption of reagents could disfavor the Fenton oxidation for potential large-scale application.

The tested reductive wastewater treatment methods are less common compared with chemical oxidation. Using appropriate reduction agents, the chemical reduction serves as a simple and economically attractive method for the destruction of a hardly biodegradable compound, such as chlorinated azo dye MB9, to non-chlorinated by-products suitable for subsequent biodegradation in common wastewater treatment plants.

Both $ZVI/H_2SO_4$ and $Al^0/H_2SO_4$ are effective azo reduction bonds, but poor HDC agents. The same inconvenience was observed in the case of $Al^0/NaOH$ action. In contrast, the heterogeneous reductant Raney Al-Ni alloy in NaOH and the homogeneous reductant $NaBH_4/Na_2S_2O_5$ alloy are very effective for both azo bond reduction and subsequent HDC.

There is still uncertainty around (i) the minimal quantity and the price of effective reductants applicable for both the azo bond reduction and effective HDC of produced chlorinated aminophenol- and aminonaphtol-based sulphonic acids and (ii) the accompanying treatment of produced by-products descended from used reductants/HDC agents.

From the consumption of the reagents point of view, the homogeneous reduction accompanied by HDC using $Na_2S_2O_4$ produced in situ (by the action of $NaBH_4/Na_2S_2O_5$) is the most effective, rapid, and simple destruction method.

An innovative heterogeneous reduction combining pretreatment with $Al^0/NaOH$ with the subsequent addition of effective HDC agents (Raney Al-Ni alloy/NaOH) is more exacting. In the case of the used Al-Ni alloy, the recycling of produced Ni-slurry was demonstrated earlier with the aim of obtaining an effective hydrodehalogenation agent.

**Supplementary Materials:** The following supporting information can be downloaded at: https://www.mdpi.com/article/10.3390/catal13030460/s1, Figure S1: The dependence of discoloration efficiency/COD removal on the pH value for the Fenton reaction using 6 mmol of $H_2O_2$ + 0.2 mmol of $FeSO_4$ per 0.05 mmol of MB9 in 200 mL of $H_2O$ (reaction time 20 h). Figure S2: LC chromatogram of LC-MS analysis of the reaction mixture obtained after the reductive degradation of MB9 using Al-Ni/NaOH. Figure S3a: LC chromatogram of LC-MS analysis of reaction mixture obtained after the reductive degradation of MB9 using $NaBH_4/Na_2S_2O_5$. Figure S3b: Qualitative LC-MS analysis of the reaction mixture obtained after the reductive degradation of MB9 using $NaBH_4/Na_2S_2O_5$. Figure S3c: Mass spectrums (EI, 70 eV) of analyzed components. Table S1: The effect of subsequently coagulation on Fenton reaction and reduction/HDC using the Al-Ni alloy. Table S2: A comparison of economic viability of MB9 removal using different oxidative/reductive agents [52–62].

**Author Contributions:** Conceptualization, T.W.; methodology, T.W.; software, B.K.; validation, B.K.; formal analysis, B.K.; investigation, B.K. and T.W.; resources, T.W.; data curation, B.K.; writing—original draft preparation, T.W.; writing—review and editing, T.W.; visualization, B.K.; supervision, T.W. All authors have read and agreed to the published version of the manuscript.

**Funding:** This research received no external funding.

**Data Availability Statement:** Not applicable.

**Acknowledgments:** The authors acknowledge the financial support for excellent technological teams from the Faculty of Chemical Technology, University of Pardubice.

**Conflicts of Interest:** The authors declare no conflict of interest.

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
