# Peer review of "A Comparison of Different Reagents Applicable for Destroying Halogenated Anionic Textile Dye Mordant Blue 9 in Polluted Aqueous Streams"

_catalysts, doi:10.3390/catal13030460_

Round 1

Reviewer 1 Report

Article entitled Comparison of different reagents applicable for destroying halogenated anionic textile dye Mordant Blue 9 in polluted aqueous streams written by Barbora Kamenická and Tomáš Weidlich and submitted to Catalysts journal as a draft no catalysts-2177589 deals with an important issue of toxic dyes removal from water.

Article is in journal’s scope. Therefore, it could be considered for publication in Catalysts journal. As English is not my native language, I am not able to assess language correctness. However, while reading I found some statements missing, confusing or unclear. Below I enclose the list of my comments.

The abstract should be completely rebuilt. The information is general and should be detailed, e.g. what methods were used specifically. Additionally, the units are not correct. Which means, for example, the use of 140 mol of sulfuric acid. It is proposed to administer the doses/concentrations of the final reagents as traditionally adopted, i.e. in relation to the volume of the solution.

Why was the CODCr used? This determination is based on a redox reaction. Same as the process used. All reagents used are factors that interfere with the correctness of the determination. How exactly did the Authors remove all disturbing influences? Why was TOC not used?

The doses of reagents proposed by the authors are very large. Was there any calculation of the economic viability of the process?

The process is very long. How do the Authors see the possibility of practical application?

Where do the compounds described in the last 2 lines of the abstract come from?

Key words are incorrect. Eg why chlorophenol?

Avoid lumped references eg 12-17.

No literature review about MB9 removal was done.

No description is given, and no literature review of the other processes used has been made.

2.1: How can you be sure that the introduced hydrogen peroxide will be consumed only for the decomposition of MB9? What's more, H2O2 is a known COD disruptor. Which explains the low COD removal.

All abbreviations should be explained at first appearance.

For the Fenton process, the optimum pH is assumed to be around 3. Why did the authors use lower ones? Was it even precisely controlled (line 104)?

Why in reaction 2 H2SO4 is used? The idea behind the Fenton reaction is a bit different.

How were the termination of the Fenton reaction performed?

How effective is the final coagulation? Was it carried out at all?

Why was the efficiency of the process not determined as a function of pH?

why the process was carried out for 20 hours, although Figure 7 shows that it should have been completed much sooner. Is there a measure for the use of hydrogen peroxide? Was hydrogen peroxide left over from the process determined?

Why ZVI/H2O2 process was not used?

The article lacks the material characteristics of ZVI, Al, Ni, alloy. It is not specified how the surface and other properties change after the process. What about the reuse of the catalyst?

Sorption related parameters of solid catalyst are not given (line 235-242).

There is no discussion of results or references to other papers in the article (except reference 18 in table 1)

Since MB9 is 50% pure, what are its other ingredients?

What was the effect of magnetic stirring on ZVI?

I have many other remarks and comments related to this article. However, based on the comments made and the overall impression, it can be concluded that:

It is not specified what the scientific novelty of this article is. On the one hand, hydrogen peroxide and the Fenton process are well known, on the other hand, the Authors indicate that the other processes have already been used.

The method of process control is very simple and with the use of parameters that are very susceptible to disturbances.

No material analysis. No emerging intermediates or toxicity identified.

The way of describing the experiment leaves much to be desired, eg doses are given in a very imprecise way, as a proportion without information on how much MB9 was used. Doses once in moles once in millimoles.

The entire materials and methods section as well as introduction should be revised.

Conclusions should be described in accordance with the convention, i.e. without references to cited literature or figures from the article.

Based on my comments and general impression I suggest to reject this article.

Author Response

Dear reviewer, enclosed please find our responses to your comments. Thank you very much for your patience.

Reviewer 2 Report

In this study, the effectiveness of various methods and catalysts in Mordant Blue 9 decolorization and oxidation were investigated. The low cost and easy accessibility of the catalysts used are highlighted. However, some issues need clarification.

1) Why did you choose a high-concentration dye solution? You did not use real wastewater.

2) Did you support the fragmentation products in the mechanisms you suggested using any instrumental method?

3) Because of the emphasis on cost, a cost calculation can be made for each method.

Author Response

Dear reviewer, thank you very much for your fruitful comments and suggestions.

Enclosed, please, find our responses.

Round 2

Reviewer 1 Report

This is my second review of this article. My first impression was negative. I described it in details my comments. However, the Authors answered all of my comments. The answer was factual, convincing and indicated a thorough thought and understanding of the subject. Suggested corrections have been applied. Second version of the manuscript is far better than the first one. After careful study of the Authors' responses and the revised version of the article, I consider that the text is worth publishing in its current form. 

Reviewer 2 Report

The explanations and arrangements made by the authors are appropriate.